# The Comparison of Fatty Acid Composition and Lipid Quality Indices in Hard Cow, Sheep, and Goat Cheeses

**DOI:** 10.3390/foods9111667

**Published:** 2020-11-15

**Authors:** Beata Paszczyk, Joanna Łuczyńska

**Affiliations:** Faculty of Food Sciences, University of Warmia and Mazury in Olsztyn, ul. Plac Cieszyński 1, 10-726 Olsztyn, Poland; jlucz@uwm.edu.pl

**Keywords:** cow, sheep, goat cheeses, *cis*9*trans*11C18:2, *trans* isomers, human health

## Abstract

This study aimed to evaluate the fatty acid composition, including the contents, of conjugated linoleic acid *cis*9*trans*11 C18:2 (CLA) and *trans* C18:1 and C18:2 isomers in hard cow, sheep, and goat cheeses found on the Polish market and to compare lipid quality indices in these cheeses. The gas chromatography method was used to determine the fatty acid profile. The study demonstrated various contents of *cis*9*trans* 11 C18:2 (CLA), *trans* C18:1, and C18:2 isomers and the values of lipid quality indices in the cheeses. Sheep and goat cheeses were richer sources of short-chain fatty acids (SCFA) (14.73 ± 2.55% and 14.80 ± 2.80%, respectively) than the cow cheeses (9.38 ± 0.87%). The cow cheeses had a significantly higher (*p* < 0.05) content of monounsaturated fatty acids (MUFA), the lowest n-6/n-3 ratio, and the highest content of fatty acids, inducing a desirable dietary effect in humans (DFA) compared to the sheep and goat cheeses. Significantly higher (*p* < 0.05) contents of polyunsaturated fatty acids (PUFA) were found in sheep cheeses. Goat cheeses had the highest n-3 PUFA content and the lowest values of the thrombogenicity index (TI) (2.67 ± 0.44) compared to the sheep and cow cheeses (3.14 ± 0.29 and 3.13 ± 0.13, respectively). The cow, sheep, and goat cheeses were characterized by similar values of the hypocholesterolemic/hypercholesterolemic (H/H) ratio. Sheep cheeses had the highest levels of *cis*9*trans*11 C18:2 (CLA) and the highest total content of *trans* C18:1 and *trans* C18:2 isomers. The research showed that sheep, cow, and goat cheeses offered various health benefits. The differences in fatty acid composition and the different values of the lipid quality indices found in the cheeses may be due to differences in both the composition of milk used to produce them and the cheese-making technology. Studies conducted by many authors have indicated that the feeding system of the ruminants has a significant impact on the quality and chemical composition of milk, as well as its applicability for cheese production.

## 1. Introduction

Milk is a good source of high-value proteins, easily digestible fat, mineral salts important for the body, a number of vitamins, and other ingredients [1]. It is secreted by the mammary gland of all mammals and contains the same nutrients, differing only in their proportions [2]. Milk fat is one of the most complex natural fats. It contains over 400 fatty acids with a different number of carbon atoms (including the even and the odd number) and, of various degrees of saturation: saturated (SFA), monounsaturated (MUFA), and polyunsaturated (PUFA), with *cis* and *trans* configuration and straight or branched carbon chains [3,4]. Some fatty acids have potential negative effects on human health [4,5]. The unfavorable feature of milk fat is the high concentration of SFAs. Excessive consumption of saturated fatty acids causes an increase in blood cholesterol levels, which is associated with an increased risk of cardiovascular disease development. The effect of SFA on serum cholesterol levels in humans depends on the length of the carbon chains of fatty acids [6,7]. According to Praagmana et al. [8], lauric acid (C12:0), myristic acid (C14:0), and palmitic acid (C16:0) increase LDL and HDL cholesterol levels, whereas stearic acid (C18:0) has neutral effects. Foods are composed of an array of saturated and unsaturated fatty acids, each of which may differentially affect lipoprotein metabolism. While conducting a study on the health effects of food, it is important to study the overall shape and composition of food, not just a single individual characteristic [9,10]. Some fatty acids present in milk fat, such as butyric acid (C4:0), oleic acid, *n*-3 and *n*-6 polyunsaturated fatty acids, *trans*-vaccenic acid (*trans*11 C18:1), and conjugated linoleic acid *cis*9*trans*11 C18:2 (CLA), have potentially positive effects on human health [11,12,13,14,15,16,17]. Milk and dairy products are the main sources of *cis*9*trans*11 C18:2 acid in man’s diet. The content of CLA in milk fat fits within a very wide range and depends on the feeding period, lactation period, breed, and individual determinants of animals. Out of these factors, the greatest significance is attributed to the period of feeding. Higher contents of this acid are found in the fat of milk from pasture feeding, whereas lower ones are in the fat of milk from stall feeding [18,19,20,21,22,23]. In the case of dairy products (cheeses or fermented drinks), the level of CLA was proven to be significantly affected not only by animal feeding but also by conditions occurring during technological processing and by the activity of the starter cultures added [24,25,26,27,28,29,30].

Cheese is an important source of essential nutrients and health-promoting compounds in the human diet, and its nutritional value depends on milk characteristics and conditions occurring during the technological processing of cheeses. Cow milk cheeses are the most popular and the most frequently consumed in Poland. Sheep and goat milk cheeses are also available on the Polish market. They are recommended for consumers allergic to cow milk. Sheep and goat milk have high nutritional value. Small differences in the composition of various milks may cause differences in the nutritional value and selected traits of the technological suitability of milk in the dairy industry [31]. Goat milk contains large amounts of short-chain fatty acids, which makes it easier to digest. Sheep milk is characterized by a higher total content of solids, fat, protein, and caseins but also by larger amounts of minerals and vitamins compared to cow and sheep milk [32,33]. The nutritional and sensory values of cheeses are influenced by many factors, including milk type and composition, starter cultures, and cheese-making technology [34].

This study aimed to determine the fatty acid composition, the contents of conjugated linoleic acid *cis*9*trans*11 C18:2 (CLA), *trans* C18:1 and C18:2 isomers, and the lipid quality indices in hard cow, sheep, and goat cheeses available on the Polish market.

## 2. Materials and Methods

### 2.1. Materials

The experimental material included commercial cow cheeses (10 samples), goat cheeses (10 samples), and sheep cheeses (10 samples). Cheese samples came from various Polish producers and were purchased in the period from September to December. All cow cheeses (Gouda, Podlaski, Edamski, Salami, Rycki, Królewski, Tylżycki, Morski, Polski, Kasztelan), two goat cheeses (Danmis, Mleczna Dolina), and two sheep cheeses (Oscypek, Ser owczy Auchan) were bought in local stores in Olsztyn (Poland). The other sheep and goat cheeses were purchased from individual farmers producing cheeses on their own farms. Goat cheeses were bought from Gospodarstwo Kozi zakątek, Gospodarstwo rolne Nad Arem, Kozia farma Złotna, Farma Serenada, Gospodarstwo hodowlane kóz mlecznych, and Gospodarstwo Tusinek, whereas sheep cheeses from Ranczo Frontiera, Gospodarstwo Pod Lipami, Gospodarstwo ekologiczne Wańczykówka, Gospodarstwo ekologiczne, “Kozia Łąka”, and Farma eko, “Owczarnia Lefevre”. All cheeses were tested during their shelf-life. All determinations were performed in duplicate.

### 2.2. Methods

#### 2.2.1. Lipid Extraction

Fat from the analyzed cheeses was extracted according to the Folch method [35]. To this end, the studied material was crushed and mixed. Approximately 3 g of sample (0.01 g) were transferred to a 100 mL beaker and homogenized (IKA Ultra-Turrax^®^T18 digital) for 1 min with 30 mL of methanol. Next, 30 mL chloroform was added, and the procedure was continued for 2 min. The prepared mixture was filtered to a 250 mL glass cylinder. The solid residue was mixed in 60 mL chloroform: methanol (2:1 *v*/*v*) and homogenized again for 3 min. The mixture was transferred to the same cylinder. An amount of 0.88% sodium chloride in water (determining ¼ volume of filtrate) was added to the total filtrate, and then, it was shaken and left overnight. The upper layer was removed, and to the lower layer, a water: methanol mixture (1:1 *v*/*v*) was added and the washing procedure was repeated. The remaining layer was filtered by filter paper with anhydrous sodium sulphate, and the solvent was later distilled.

#### 2.2.2. Determination of the Fatty Acid Profile

Fatty acids were converted into the corresponding fatty acid methyl esters (FAME) according to the IDF standard method (ISO 15884:2002) [36]. For this purpose, 50 mg of extracted fat was placed inside a sealed ampule of a capacity of 7 mL, and 2.5 mL hexane was then added to dissolve it. The ampoule content was shaken vigorously until the fat was completely dissolved. Next, 0.1 mL 2M methanolic KOH solution was added. The ampoule was vigorously mixed in the vortex mixer for 1 min and then left for 5 more minutes at room temperature (ca. 20 °C). After that time, 0.25 g of sodium hydrogen sulphate-monohydrate was then the mixture was added and spun for 3 min (approximately 3000 spins/minute). The top layer of prepared methyl esters was taken for gas chromatographic (GC) analysis. Chromatographic separation was performed using a Hewlett-Packard 6890 gas chromatograph (Műnster, Germany) with a flame-ionization detector (FID) and a capillary column with a length of 100 m and internal diameter of 0.25 mm. The liquid phase was CP Sil 88 (Chrompack, Middelburg, The Netherlands) and the film thickness was 0.2 μm. The conditions of separation were as follows: carrier gas: helium, gas flow 1.5 mL/min, column temperature from 60 °C (for 1 min) to 180 °C, Δt = 5 °C/min, and the detector temperature −250 °C and injector temperature 225 °C. Sample injection volume was 0.4 μL (split: 50:1). Methyl esters of fatty acids were identified according to their retention times, which were then compared with the retention time of the methyl esters of fatty acids of reference milk fat (BCR Reference Materials) of CRM 164 symbol. For the identification of positional *trans* C18:1 isomers, the standards of methyl esters of these isomers (Sigma-Aldrich, Germany) were used. For the identification of positional *trans* C18:2 isomers, a mixture of standards of C18:2 isomers (Supelco, Bellefonte, PA, USA) was used, and for the identification of *cis*9*trans*11 C18:2 (CLA) isomer, a mixture of CLA methyl esters (Sigma-Aldrich, Germany). Amounts of fatty acids were expressed as a weight percentage of total methyl esters of fatty acid.

#### 2.2.3. The Lipid Quality Indices

The AI and TI indices were calculated using the following formulae [6,37]:

Index of Atherogenicity (AI):AI = (C12:0 + (4 × C14:0) + C16:0)/(Σn-3 PUFA + Σn-6 PUFA + Σ MUFA)

Index of Thrombogenicity (TI):TI = (C14:0 + C16:0 + C18:0)/((0.5 × C18:1) + (0.5 × other MUFA) + (0.5 × Σn-6 PUFA) + (3 × Σn-3 PUFA) + Σn-3 PUFA/Σn-6PUFA)

Hypocholesterolemic Fatty Acids (DFA) was calculated according to Osmari et al. [37]:DFA = UFA + C18:0

Hypercholesterolemic Fatty Acids (OFA)
OFA = C12:0 + C14:0 + C16:0

The ratio of hypocholesterolemic and hypercholesterolemic fatty acids (H/H) was calculated according to the fatty acids composition using the following formulae [38]:H/H = (C18:1n-9 + C18:2n-6 + C18:3n-3)/(C12:0 + C14:0 + C16:0)

#### 2.2.4. Statistical Analysis

The statistical analysis was carried out using STATISTICA ver.13.1 software (Statsoft, Kraków, Poland) [39]. To calculate the significance of differences, the one-way analysis of variance (ANOVA). The significance level of *p* < 0.05 was used. Differences between mean values were evaluated with Duncan’s test.

## 3. Results and Discussion

### 3.1. Fatty Acid Composition and Lipid Quality Indices in Cheeses

The fatty acid composition of fat extracted from the cow, sheep, and goat cheeses is presented in Table 1. The total content of fatty acid groups and lipid quality indices in the cheeses are presented in Table 2. The results presented in Table 2 indicate that saturated fatty acids (SFA) predominated in the fat extracted from cow, sheep, and goat cheeses. Sheep cheeses had significantly lower (*p* < 0.05) contents of these acids (56.61 ± 1.95%) than cow cheeses (59.41 ± 0.91%). In goat cheeses, the content of SFA was 58.08 ± 3.27%. Palmitic acid (C16:0), myristic acid (C14:0), and stearic acid (C18:0) were found to be the major SFA in all cheeses (Table 1). According to Kawęcka et al. [40], the content of SFA in sheep cheeses was lower than in cow and goat cheeses. Naturally high levels of saturated fatty acids in milk and dairy products are often associated with the adverse effects of these products’ consumption and with the development of many diseases, including cardiovascular disease, type 2 diabetes mellitus, obesity, and cancer. Recent findings have indicated that the link between SFA and the incidence of these diseases may be less evident than previously assumed. Foods are composed of an array of saturated and unsaturated fatty acids, each of which may differentially affect lipoprotein metabolism and contribute to significant quantities of other nutrients, which can affect the risk of development of many diseases [9,10]. Sheep and goat cheeses contained significantly higher (*p* < 0.05) quantities of SCFA (14.73 ± 2.55% and 14.80 ± 2.80%, respectively) than cow cheeses (9.38 ± 0.89%) (Table 2). Considering this group of fatty acids, the highest contents of C4:0 acid were found in the fat extracted from cow cheeses. Short-chain fatty acids are important in promoting human health. For instance, butyric acid has been proved to exhibit anti-inflammatory activity and to prevent the progression of colorectal cancer and mammary cancer [41,42]. In the fat from goat and sheep cheeses, the major SCFA was capric acid (C10:0), with its content exceeding 7% of the total fatty acids (Table 1).

The contents of MUFAs in the fat extracted from goat and sheep cheeses were similar, while their significantly higher (*p* < 0.05) contents were found in cow cheese (Table 2). In all analyzed cheese samples, oleic acid (C18:1 *cis*9) was the major acid of the MUFA group (Table 1). It has been documented to exhibit anti-cancer and anti-atherogenic properties and, therefore, is useful in an everyday diet [41]. The study showed that the fat extracted from sheep cheeses contained significantly higher (*p* < 0.05) amounts of PUFA (4.36 ± 0.25%) than cow and goat cheeses (3.31 ± 0.33% and 3.49 ± 0.47%, respectively) (Table 2). In cow cheeses analyzed by Prandini et al. [43], the SFA content ranged from 65.23% to 68.52%, that of MUFA ranged from 27.90% to 31.19%, and that of PUFA ranged from 3.48% to 4.17%. Goat and ewe cheeses analyzed by these authors contained 72.92% and 67.69% of SFA, respectively. The contents of MUFA and PUFA were at 23.03% and 4.04% in goat cheeses, and at 26.83% and 5.48% in ewe cheeses, respectively. In sheep cheeses analyzed by Milewski et al. [44], the contents of MUFA and PUFA were lower than in the cheese analyzed in the present study, i.e., 22.04% and 3.09%, respectively. In goat cheeses analyzed by these authors, the MUFA content was higher than in our study (26.34%), while the PUFA content was lower (2.85%).

The study showed that the fat extracted from sheep and cow cheeses had significantly higher (*p* < 0.05) contents of *n*-3 acids and lower *n*-6/*n*-3 ratio compared to goat cheeses (Table 2). In turn, the fat extracted from sheep and goat cheeses had significantly higher contents of *n*-6 PUFAs compared to cow cheeses. Proportions of specific groups of fatty acids in products are of special importance from the nutritional perspective. Excessive amounts of n-6 polyunsaturated fatty acids (PUFA) and a very high n-6/n-3 ratio, as is found in today’s Western diets, promote the pathogenesis of many diseases, including cardiovascular disease, cancer, and inflammatory and autoimmune diseases, whereas increased levels of n-3 PUFA (a low *n*-6/*n*-3 ratio) exert suppressive effects [45,46,47,48]. In the present study, the n-6/n-3 PUFA ratio was the lowest in cow cheeses and reached 3.37 (Table 2). In sheep cheeses, it was 4.62, while goat cheeses were characterized by its significantly higher (*p* < 0.05) value (6.43). In addition, Aguilar et al. [49] and Kawęcka et al. [40], who compared fatty acid profiles in cheeses made from sheep, cow, and goat milk, demonstrated that the *n*-6/*n*-3 ratio was higher in goat than in sheep and cow cheeses. In fresh goat cheeses analyzed by Cossignani et al. [50], the *n*-6/*n*-3 ratio was 7.0, and in goat semi-hard cheeses, it was 3.3. According to Hirigoyen et al. [51], the *n*-6/*n*-3 ratio was 3.29 in “Colonia” cheeses produced from cow milk in autumn and 4.47 in those produced in the spring. In sheep and goat cheeses analyzed by Milewski et al. [44], the *n*-6/*n*-3 ratio was lower (reading 3.42 and 2.94, respectively) compared to the sheep and goat cheeses analyzed in the presented study. The study also showed that the content of desirable hypocholesterolemic fatty acids (DFAs) was the highest in cow cheeses (41.78 ± 1.17). Their significantly lower (*p* < 0.05) contents were found in sheep and goat cheeses. While comparing the fatty acid profile in cheeses made from sheep, cow, and goat milk, Kawęcka et al. [40] demonstrated a higher DFA content in cow and sheep cheeses than in goat cheeses. The data presented in Table 2 show that the cheeses had a similar content of fatty acids with undesired hypercholesterolemic effect (OFA). In sheep cheeses analyzed by Milewski et al. [44], the DFA content was 35.83 and that of OFA was 64.18. In goat cheeses analyzed by these authors, the DFA and OFA contents were at 40.47 and 59.57, respectively.

In the presented study, the AI value in sheep cheeses was 2.85 ± 0.30, which was significantly higher (*p* < 0.05) than in goat cheeses (1.78 ± 0.32) and cow cheeses (1.63 ± 0.06). The TI value was the highest in sheep and cow cheeses, i.e., 3.14 ± 0.29 and 3.13 ± 0.13, respectively. A significantly lower (*p* < 0.05) value (2.67 0.44) was found in goat cheeses (Table 2.). Lower AI values and higher TI values in sheep and goat cheeses were found by Milewski et al. [44]. In turn, Aguilar et al. [49] reported higher AI and TI values in goat cheeses than in sheep and cow cheeses. Cossignani et al. [50] reported that the AI value was 2.9 in goat milk, 2.7 in fresh cheese samples, and 2.4 in semi-hard cheese samples. Hirigoyen et al. [51] reported that AI and TI values calculated for “Colonia” cheese produced from cow milk in the autumn and in the spring were 2.21 and 2,84, respectively for both seasons. According to Ulbricht and Southgate [6], the AI and TI indices might better characterize the atherogenic and thrombogenic potential of the diet than the PUFA/SFA ratio. The atherogenicity index (AI) and thrombogenicity index (TI) take account of the different effects that single fatty acids might have on human health and, in particular, on the probability of increasing the incidence of pathogenic phenomena, such as atheroma and/or thrombus formation. The AI value indicates the relationship between the sum of the main saturated fatty acids and that of the main classes of unsaturated fatty acids. In contrast, the TI value shows the tendency for blood clot formation in the blood vessels. This is defined as the relationship between the prothrombogenic (SFA) and the antithrombogenic fatty acids (MUFA *n*-3 and *n*-6 PUFA). Thus, the higher the AI, the more atherogenic dietary components there are. The low AI value indicates that milk and milk products could provide protection against coronary heart diseases.

The cow, sheep, and goat cheeses were characterized by similar values of H/H (Table 2). The H/H ratio is related to the functional activity of fatty acids in the metabolism of lipoproteins for plasma cholesterol transport and to the risk of cardiovascular disease development. Higher values of this ratio are desirable [52].

### 3.2. The Contents of CLA and Trans C18:1 and Trans C18:2 Fatty Acids in Cheeses

Table 3 shows the contents of *cis*9*trans*11 C18:2 (CLA) and *trans* isomers of C18:1 and C18:2 acids in the fat extracted from cow, sheep, and goat cheeses. The data shown in Table 3 indicate that the cheeses had different contents of *cis*9*trans*11 C18:2 (CLA) acid, which is the major CLA isomer in food. In milk and dairy products, it accounts for more than 80–90% of the total CLA content and is thought to exhibit high biological activities [53,54]. In the fat extracted from sheep cheeses, the CLA content ranged from 0.49 to 1.52% of the total fatty acids. Less variation was observed in the cow and goat cheeses (Table 2). The average content of CLA in the total fatty acid composition in sheep cheeses (0.75 ± 0.31%) was significantly higher (*p* < 0.05) that in goat cheeses (0.48 ± 0.10%). In cow cheeses, the average content of CLA was 0.65 ± 0.12% of the total fatty acids (Table 2). In German cow cheeses analyzed by Fritsche and Steinhart [55], the content of CLA in the total fatty acid composition ranged from 0.40 to 1.70%, while in goat and sheep cheeses it was 0.50 and 1.01%, respectively. Zlatanos et al. [56] report that the content of CLA in Greek hard cheeses with short ageing time and hard cheeses with long ageing time ranged from 0.5 to 1.1% and from 0.5 to 1.9%, respectively. The content of *cis*9*trans*11 C18:2 acid in Turkish hard cheeses produced from cow milk ranged from 0.44 to 3.01% of the total fatty acids [57]. Grega et al. [58] reported that the CLA content ranged from 0.20 to 0.95% in the fat from commercial cheeses produced in winter and from 0.61 to 1.57% in the fat of cheeses from the summer period. According to Żegarska et al. [59], the content of *cis*9*trans*11 C18:2 acid in hard commercial cheeses made from cow milk, purchased in February and March, ranged from 0.48 to 1.68% of the total fatty acids. In hard cheeses bought in October and November, the CLA content ranged from 0.97 to 1.46% of the total fatty acids. In sheep cheeses analyzed by Milewski et al. [44], the content of *cis*9*trans*11 C18:2 (CLA) reached 1.09%. In goat cheeses it was 0.73%. In Italian and French commercial cheeses obtained from milk of different ruminant species (cow, goat and sheep) analyzed by Prandini et al. [60], sheep cheeses had the highest CLA levels.

Monounsaturated fatty acids with 18 carbon atoms are the most prominent *trans* fatty acids (TFA) in the human diet. The average total content of *trans* isomers of C18:1 in the fat extracted from goat cheeses (2.24 ± 0.58%) was significantly lower (*p* < 0.05) than in the fat extracted from cow and sheep cheeses (2.89 ± 0.33% and 3.27 ± 0.71%, respectively) (Table 3). Considering the group of *trans* C18:1, all cheese samples contained the highest amounts of *trans*10 + *trans*11 isomers (Table 3). In the fat extracted from cow and sheep cheeses, the content of these isomers was 1.81 ± 0.30% and 2.05 ± 0.67%, respectively, and it was significantly higher (*p* < 0.05) than in the fat extracted from goat cheese (1.23 ± 0.35%). *Trans*11 C18:1 (vaccenic acid, VA) is the main TFA in milk fat, and its contents can be influenced by the animal feeding system [61,62]. Under conventional ruminant diets, milk contains around 40–50% of VA in total C18:1 TFA, whereas the contents of *trans*9 and *trans*10 C18:1 are considerably lower (5% and 10% on average, respectively) [63,64]. *Trans*6, *trans*9, and *trans*12 C18:1 isomers were at similar levels in all cheeses. In the fat extracted from goat cheeses, the content of *trans*16 isomer was significantly lower (*p* < 0.05) than in the other cheeses (Table 3). The study conducted by Aguilar et al. [49] to compare fatty acid profiles in cheeses made from sheep, cow, and goat milk demonstrated a higher content of C18:1 *trans* isomers in sheep milk compared with cow and goat milk cheeses. In sheep cheeses, the content of these isomers was at 3.77 ± 1.21%, and in cow and goat cheeses, it reached 3.36 ±1.65% and 2.47 ±1.26%, respectively. The higher levels of C18:1 *trans* isomers in the cheeses may be due to the different feeding systems. Cheeses made from milk obtained from forage-fed animals with added oilseeds or fresh grass may contain more of these isomers. According to Chilliard et al. [65], when pasture inclusion increases in diet, linear increases in C18:3 *n*-3, *trans*11 C18:1, and *cis*9*trans*11 C18:2 and decreases in C10:0–C16:0 are observed.

The highest content of *trans* C18:2 isomers was found in sheep cheeses (0.82 ± 0.19% of the total fatty acids), whereas cow and goat cheeses had significantly lower (*p* < 0.05) contents of these isomers (Table 3).

## 4. Conclusions

Data obtained in the study indicate that the contents of the fatty acids CLA, *trans* C18:1, and C18:2 isomers, as well as the lipid quality indices, varied in the cheeses examined. The sheep and goat cheeses were richer sources of SCFA than cow cheeses. In turn, cow cheeses were characterized by the highest content of MUFA, the lowest *n*-6/*n*-3 ratio, and the highest content of DFA. Higher contents of CLA and PUFAs were found in sheep cheeses. Goat cheeses were characterized by the highest n-3 PUFA and the lowest TI index value, compared to the other cheeses. The conducted research shows that sheep, cow, and goat cheeses offer various health benefits. The differences in the composition of fatty acids and the different values of lipid quality indices in the cheeses may be due to the differences in both the composition of milk used to produce them and cheese-making technology. Studies carried out by many authors have indicated that the feeding system of the ruminants has a significant impact on the quality and chemical composition of milk, as well as on its applicability for cheese production.

## Figures and Tables

**Table 1 foods-09-01667-t001:** Fatty acid composition in fat extracted from cheeses (% of total fatty acids).

Fatty Acid	Cow Cheeses			Sheep Cheeses			Goat Cheeses		
n	10			10			10		
	Mean	±SD	Min–Max	Mean	± SD	Min–Max	Mean	±SD	Min–Max
C4:0	3.04 ^a^	0.40	2.19–3.34	2.83 ^a^	0.41	2.49–3.39	2.17 ^b^	0.86	1.13–3.62
C6:0	2.06 ^b^	0.33	1.83–2.31	2.42 ^a^	0.32	1.69–2.26	2.29 ^a^	0.32	1.79–2.80
C8:0	1.31 ^b^	0.11	1.05–1.40	2.33 ^a^	0.48	1.51–2.78	2.39 ^a^	0.57	1.54–3.09
C10:0	2.97 ^b^	0.12	2.80–3.18	7.14 ^a^	1.73	3.52–8.64	7.95 ^a^	2.81	3.78–10.90
C10:1	0.30 ^a^	0.02	0.27–0.32	0.27 ^a^	0.03	0.24–0.33	0.30 ^a^	0.10	0.19–0.49
C11:0	0.05 ^c^	0.01	0.04–0.06	0.08 ^b^	0.02	0.04–0.10	0.11 ^a^	0.01	0.09–0.13
C12:0	3.22 ^c^	0.52	1.86–3.58	4.40 ^b^	0.52	3.25–4.99	5.19 ^a^	0.78	4.34–6.44
C12:1	0.04 ^a^	0.00	0.04–0.05	0.04 ^a^	0.01	0.02–0.06	0.03 ^b^	0.01	0.02–0.04
C13:0 *iso*	0.07 ^a^	0.00	0.07–0.08	0.04 ^b^	0.01	0.03–0.08	0.07 ^a^	0.03	0.03–0.13
C13:0	0.10 ^b^	0.01	0.08–0.12	0.09 ^b^	0.03	0.06–0.15	0.13 ^a^	0.03	0.07–0.18
C14:0 *iso*	0.20 ^a^	0.08	0.11–0.28	0.14 ^a^	0.05	0.10–0.26	0.11 ^a^	0.02	0.09–0.15
C14:0	11.39 ^a^	0.27	11.03–11.67	11.48 ^a^	0.90	10.37–12.66	11.86 ^a^	1.38	9.66–14.72
C15:0 *iso*	0.43 ^a^	0.12	0.28–0.55	0.30 ^a^	0.08	0.23–0.48	0.22 ^a^	0.04	0.17–0.29
C15:0 *aiso*	0.54 ^a^	0.04	0.50–0.62	0.50 ^a^	0.10	0.41–0.68	0.45 ^b^	0.12	0.33–0.69
C14:1	0.97 ^a^	0.07	0.86–1.07	0.30 ^b^	0.26	0.16–0.98	0.53 ^b^	0.55	0.32–1.51
C15:0	1.20 ^a^	0.05	1.14–1.31	1.11 ^a^	0.14	0.99–1.42	1.18 ^a^	0.27	0.86–1.68
C16:0 *iso*	0.31 ^a^	0.02	0.27–0.33	0.30 ^a^	0.08	0.25–0.49	0.28 ^a^	0.05	0.21–0.38
C16:0	29.70 ^a^	1.03	27.71–31.02	26.75 ^a^	1.92	24.52–30.99	28.66 ^a^	1.69	25.26–32.56
C17:0 *iso*	0.43 ^b^	0.05	0.37–0.54	0.48 ^a^	0.10	0.37–0.60	0.39 ^b^	0.04	0.32–0.44
C17:0 *aiso*	0.23 ^b^	0.02	0.19–0.27	0.29 ^a^	0.04	0.24–0.36	0.28 ^a^	0.05	0.18–0.33
C16:1	1.63 ^a^	0.20	1.42–2.00	1.01 ^b^	0.36	0.66–1.86	1.15 ^b^	0.10	1.63–2.16
C17:0	0.72 ^a^	0.03	0.67–0.76	0.70 ^a^	0.09	0.62–0.73	0.70 ^a^	0.12	0.59–0.94
C17:1	0.27 ^a^	0.01	0.26–0.29	0.25 ^a^	0.05	0.21–0.35	0.24 ^a^	0.03	0.20–0.28
C18:0	10.55 ^a^	0.51	9.73–11.52	9.50 ^b^	0.69	8.30–10.43	8.22 ^c^	1.36	6.74–20.24
*trans*6 − *trans*9 C18:1	0.45 ^a^	0.02	0.42–0.48	0.48 ^a^	0.07	0.39–0.58	0.45 ^a^	0.12	0.30–0.64
*trans1*0 + *trans*11 C18:1	1.81 ^a^	0.31	1.33–2.22	2.06 ^a^	0.69	1.19–3.69	1.23 ^b^	0.36	0.93–1.84
*trans* 12 C18:1	0.30 ^a^	0.02	0.27–0.33	0.37 ^a^	0.09	0.25–0.49	0.30 ^a^	0.09	0.17–0.46
*cis*9 C18:1	20.60 ^a^	0.40	20.12–21.19	18.29 ^b^	1.39	15.58–19.92	18.15 ^b^	1.45	16.88–21.58
*cis*11 C18:1	0.73 ^a^	0.05	0.68–0.86	0.50 ^c^	0.04	0.45–0.56	0.62 ^b^	0.14	0.46–0.89
*cis*12 C18:1	0.27 ^a^	0.04	0.20–0.30	0.32 ^a^	0.13	0.11–0.46	0.28 ^a^	0.10	0.18–0.46
*cis*13 C18:1	0.11 ^b^	0.01	0.10–0.12	0.08 ^a^	0.01	0.05–0.09	0.07 ^a^	0.02	0.05–0.10
*trans*16 C18:1	0.33 ^a^	0.02	0.28–0.35	0.36 ^a^	0.06	0.26–0.49	0.26 ^b^	0.06	0.19–0.39
C19:0	0.19 ^a^	0.02	0.17–0.21	0.24 ^a^	0.04	0.21–0.35	0.16 ^c^	0.02	0.14–0.20
*cis*9*trans*13 C18:2	0.19 ^c^	0.02	0.16–0.23	0.31 ^a^	0.05	0.27–0.43	0.23 ^b^	0.03	0.18–0.28
*cis*9*trans*12 C18:2	0.18 ^b^	0.01	0.17–0.20	0.26 ^a^	0.03	0.22–0.32	0.21 ^b^	0.04	0.16–0.29
*trans*9cis12 C18:2	0.03 ^a^	0.02	0.01–0.05	0.03 ^a^	0.03	0.01–0.09	0.14 ^a^	0.05	0.07–0.23
*trans*11*cis*15 C18:2	0.22 ^a^	0.08	0.12–0.32	0.22 ^a^	0.14	0.06–0.50	0.14 ^b^	0.07	0.05–0.31
*cis*9*cis*12 C18:2	1.55 ^b^	0.08	1.42–1.66	2.22 ^a^	0.60	1.00–2.81	2.07 ^a^	0.36	1.65–2.70
C20:0	0.16 ^b^	0.01	0.15–0.18	0.25 ^a^	0.10	0.14–0.49	0.17 ^b^	0.03	0.12–0.22
C20:1	0.11 ^a^	0.01	0.11–0.12	0.02 ^b^	0.03	0.01–0.11	0.05 ^b^	0.05	0.02–0.10
*cis*9*cis*12*cis*15 C18:3	0.48 ^b^	0.11	0.33–0.60	0.56 ^a^	0.19	0.37–0.75	0.34 ^b^	0.07	0.22–0.51
*cis*9*trans*11 C18:2 (CLA)	0.65 ^a,b^	0.12	0.47–0.83	0.75 ^a^	0.32	0.49–1.51	0.48 ^b^	0.10	0.37–0.64

n—number of samples; Mean—mean value; SD—standard deviation; Min—minimum value, Max—maximum value, ^a,b,c^—values denoted in rows by different letters indicate statistically significant differences (*p* < 0.05).

**Table 2 foods-09-01667-t002:** Sum of fatty acids (% of total fatty acids) and nutritional indices in fat extracted from cow, sheep and goat cheeses.

	Cow Cheeses				Sheep Cheeses				Goat Cheeses			
n	10				10				10			
	Mean	SD	Min–Max		Mean	SD	Min–Max		Mean	SD	Min–Max	
Σ SCFA ^1^	9.38 ^b^	0.87	7.64	10.11	14.73 ^a^	2.55	10.43	17.15	14.80 ^a^	2.80	10.55	18.45
Σ SFA ^2^	59.41 ^a^	0.91	58.37	60.81	56.61 ^b^	1.94	54.60	60.65	58.08 ^a,b^	3.27	52.72	61.64
Σ MUFA ^3^	27.92 ^a^	0.55	27.11	28.56	24.36 ^b^	2.08	21.05	28.52	23.66 ^b^	2.90	21.41	30.56
Σ PUFA ^4^	3.31 ^b^	0.33	2.84	3.73	4.36 ^a^	0.25	3.92	4.65	3.49 ^b^	0.47	2.99	4.24
n-3	0.48 ^a^	0.11	0.33	0.60	0.56 ^a^	0.19	0.37	0.75	0.34 ^b^	0.07	0.22	0.51
n-6	1.55 ^b^	0.08	1.42	1.66	2.22 ^a^	0.60	1.00	2.81	2.07 ^a^	0.36	1.65	2.45
n6/n3	3.37 ^b^	0.81	2.77	4.97	4.62 ^b^	2.22	1.14	7.43	6.43 ^a^	1.98	4.68	11.14
UFA ^5^	31.23 ^a^	0.77	29.95	32.02	28.72 ^b^	2.17	25.14	32.98	27.15 ^b^	3.06	24.36	34.34
DFA ^6^	41.78 ^a^	1.17	40.08	43.51	38.21 ^b^	2.43	33.79	42.18	35.37 ^c^	4.20	31.30	44.55
OFA ^7^	48.86 ^a^	1.27	46.85	50.02	47.11 ^a^	2.60	44.17	52.35	49.87 ^a^	4.43	42.51	53.73
AI ^8^	1.63 ^b^	0.06	1.56	1.74	2.85 ^a^	0.30	2.48	3.44	1.78 ^b^	0.32	1.12	2.14
TI ^9^	3.13 ^a^	0.13	2.94	3.36	3.14 ^a^	0.29	2.62	3.63	2.67 ^b^	0.44	1.58	3.13
H/H ^10^	0.55 ^a^	0.02	0.52	0.60	0.55 ^a^	0.05	0.52	0.63	0.52 ^a^	0.10	0.44	0.75

n—number of samples, Mean—mean value, SD—standard deviation, Min—minimum value, Max—maximum value, ^a,b,c^—values denoted in rows by different letters indicate statistically significant differences (*p* < 0.05), ^1^ Σ SCFA—all short-chain fatty acids (C4:0, C6:0, C8:0, C10:0); ^2^ Σ SFA—all saturated fatty acids; ^3^ Σ MUFA—all monounsaturated fatty acids; ^4^ Σ PUFA—all polyunsaturated fatty acids, ^5^ UFA—all unsaturated fatty acids (Σ MUFA + Σ PUFA), ^6^ DFA—hypocholesterolemic fatty acids (Σ UFA + C18:0); ^7^ OFA—hypercholesterolemic fatty acids (Σ SFA-C18:0), ^8^ AI—Index of Atherogenicity; ^9^ TI—Index of Thrombogenicity, ^10^ H/H—hypocholesterolemic/hypercholesterolemic ratio.

**Table 3 foods-09-01667-t003:** CLA and t*rans* fatty acids in fat from analyzed cheeses (% of total fatty acids).

	Cow Cheeses			Sheep Cheeses			Goat Cheeses		
n	10			10			10		
	Mean	SD	Min–Max	Mean	SD	Min–Max	Mean	SD	Min–Max
*cis*9*trans*11 C18:2 (CLA)	0.65 ^a,b^	0.12	0.46–0.85	0.75 ^a^	0.31	0.49–1.52	0.48 ^b^	0.10	0.36–0.61
*trans*6 − *trans*9 C18:1	0.45 ^a^	0.02	0.41–0.48	0.48 ^a^	0.08	0.35–0.58	0.44 ^a^	0.12	0.29–0.60
*trans1*0 + *trans*11 C18:1	1.81 ^a^	0.30	1.31–2.29	2.05 ^a^	0.67	1.16–3.71	1.23 ^b^	0,35	0.93–1.91
*trans* 12 C18:1	0.30 ^a^	0.03	0.27–0.34	0.37 ^a^	0.09	0.24–0.49	0.30 ^a^	0.30	0.16–0.51
*trans*16 C18:1	0.32 ^a^	0.02	0.28–0.35	0.36 ^a^	0.06	0.25–0.44	0.26 ^b^	0.06	0.18–0.34
Σ *trans* C18:1	2.89 ^a^	0.33	2.58–3.27	3.27 ^a^	0.71	2.10–4.74	2.24 ^b^	0.58	1.74–3.04
*cis*9*trans*13 C18:2	0.19 ^c^	0.02	0.16–0.23	0.31 ^a^	0.05	0.26–0.43	0.23 ^b^	0.04	0.17–0.32
*cis*9*trans*12 C18:2	0.18 ^b^	0.02	0.14–0.24	0.26 ^a^	0.04	0.18–0.34	0.21 ^b^	0.05	0.14–0.29
*trans*9*cis*12 C18:2	0.03 ^a^	0.02	0.01–0.06	0.03 ^a^	0.03	0.01–0.11	0.01 ^a^	0.02	0.01–0.07
*trans*11*cis*15 C18:2	0.22 ^a^	0.08	0.12–0.32	0.22 ^a^	0.14	0.06–0.50	0.14 ^a^	0.07	0.05–0.31
Σ *trans* C18:2	0.63 ^b^	0.09	0.50–0.75	0.82 ^a^	0.19	0.63–1.18	0.60 ^b^	0.08	0.05–0.73

Mean—mean value, SD—standard deviation; ^a,b,c^—values denoted in rows by different letters indicate statistically significant differences (*p* < 0.05).

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
