# Peer review of "The Comparison of Fatty Acid Composition and Lipid Quality Indices in Hard Cow, Sheep, and Goat Cheeses"

_foods, 2020, doi:10.3390/foods9111667_

Round 1

Reviewer 1 Report

Brief summary:

The objective of this research was to evaluate the fatty acid composition, including the concentrations of CLA cis-9, trans-11; C18:1 trans; and C18:2 isomers, of hard bovine, ovine, and caprine cheeses available in Poland. Moreover, major lipid quality indices were also determined and compared in the samples tested. All things considered, cow’s, sheep’s, and goat’s cheeses were shown to have differing health-promoting properties. The authors suggested that all three types of cheese be included in the human diet.

Broad comments:

The novelty value of this study is limited. In addition, sample sizes were not large enough to produce very robust results, and the cheese samples tested were collected at a specific time of the year (from September to December). The latter issue is problematic because season is known to greatly affect the composition, including the FA composition, of milk and cheese. Nevertheless, the methods and reagents used are mostly described in sufficient detail, and the data presented appear to support the conclusions reached. However, the English presentation needs to be improved.

Specific comments:

  • The conclusions of the research are missing from the Abstract.
  • Lines 41–42, 141, and 146: There is some contradiction here. Are C16:0 and C18:0 neutral, hypocholesterolemic, or hypercholesterolemic then?
  • Lines 48 and 55: A maximum of three citations are sufficient to provide evidence for a point being made. More than 10 references (i.e., 11-21 and 28-40) are both excessive and unnecessary.
  • Subsection 2.1:
    • Please specify if all the cheese samples tested were manufactured in Poland or they were partly imported from abroad.
    • The name of commercial hard cheese products should be indicated. Similarly, it would be reasonable to provide data about the major composition of cheese samples.
    • Were the cheese milk batches produced at about the same time of the year? Along with other factors (e., breed, stage of lactation, parity, management or production system, feeding, geographical location, year, etc.), season can largely influence the composition of milk and cheese. Instead of (or in addition to) the date of purchase, the date of manufacture should be given.
  • I failed to notice the results of statistical analyses in Table 1.
  • Lines 218–222: It is not clear whether authors argue for or against n-6 FA. Let me call their attention to the following information: “…humankind is known to have evolved on a diet with a ratio of n-6 to n-3 FA of close to 1:1. However, Western societies today are characterized by increased n-6 and trans FA and decreased n-3 FA intakes... In other words, the current nutritional environment in developed countries largely differs from that on which our genetic patterns were established roughly forty millennia ago... Therefore, a significant reduction in n-6 FA is desirable because excessive doses of n-6 PUFA and extremely high n-6 to n-3 FA ratios (i.e., 15–17:1) promote the pathogenesis of a wide range of chronic diseases, including inflammatory bowel disease, rheumatoid arthritis, cancer, and cardiovascular and autoimmune diseases; whereas decreased n-6:n-3 ratios have been shown to exert suppressive effects” [Toth et al. (2019) Int. Dairy J., 99, 104552].

Author Response

Response to Reviewer 1 Comments

Thank you very much for  reviews of the manuscript. Please find below the answers to the suggested remarks. All changes have been included in the manuscript.

Reviewer:  The conclusions of the research are missing from the Abstract

Response: As suggested by the Reviewer, the part “Abstract” has been corrected.

Reviewer:    Lines 41–42, 141, and 146: There is some contradiction here. Are C16:0 and C18:0 neutral, hypocholesterolemic, or hypercholesterolemic then?

Response: Thank You for this comment.  As suggested by the Reviewer, this sentence has been changed to:

“The effect of SFA on serum cholesterol levels in humans depends on the length of the carbon chains of fatty acids [6, 7]. According to Praagmana et al. [8], lauric acid (C12:0), myristic acid (C14:0), and palmitic acid (C16:0) increase LDL and HDL cholesterol levels, whereas stearic acid (C18:0) has neutral effects.”

Reviewer:  Lines 48 and 55: A maximum of three citations are sufficient to provide evidence for a point being made. More than 10 references (i.e., 11-21 and 28-40) are both excessive and unnecessary.

Response: As suggested by the Reviewer, this has been changed. 

Reviewer:  Please specify if all the cheese samples tested were manufactured in Poland or they were partly imported from abroad. The name of commercial hard cheese products should be indicated. Similarly, it would be reasonable to provide data about the major composition of cheese samples.

Response: As suggested by the Reviewer, the part “Materials” has been corrected.

Cheese samples came from various Polish producers and were purchased in the period from September to December. All cow cheeses (Gouda, Podlaski, Edamski, Salami, Rycki, Królewski, Tylżycki, Morski, Polski, Kasztelan), two goat cheeses (Danmis, Mleczna Dolina) and three sheep cheeses (Oscypek, Ser owczy Auchan) were bought in local stores in Olsztyn (Poland). The other sheep and goat cheeses were purchased from individual farmers producing cheeses on their own farms. Goat cheeses were bought from: Gospodarstwo Kozi zakątek, Gospodarstwo rolne Nad Arem, Kozia farma Złotna, Farma Serenada, Gospodarstwo hodowlane kóz mlecznych, and Gospodarstwo Tusinek, whereas sheep cheeses from: Ranczo Frontiera, Gospodarstwo Pod Lipami, Gospodarstwo ekologiczne Wańczykówka, Gospodarstwo ekologiczne “Kozia Łąka", and Farma eko “Owczarnia Lefevre". All cheeses were tested during their shelf-life.

The paper does not provide the main composition of the cheese samples because it has not been determined. As presented, for the purpose of this study, we determined only the fatty acid profile, the content of conjugated linoleic acid cis9trans11 C18:2 (CLA), trans C18:1 and C18:2 isomers and lipid quality indices.

Reviewer: Were the cheese milk batches produced at about the same time of the year? Along with other factors (e., breed, stage of lactation, parity, management or production system, feeding, geographical location, year, etc.), season can largely influence the composition of milk and cheese. Instead of (or in addition to) the date of purchase, the date of manufacture should be given.

Response: I agree with the Reviewer that many factors can affect the fatty acid profile of cheeses. The season of milk and cheese production and the conditions used in the production process can have a big impact. The aim of this study, however, was to evaluate the fatty acid composition of sheep, cow and goat cheeses available on the market in the same period.

Reviewer: I failed to notice the results of statistical analyses in Table 1.

Response: As suggested by the Reviewer, the data in Table 1 have been supplemented.

Reviewer: Lines 218–222: It is not clear whether authors argue for or against n-6 FA. Let me call their attention to the following information: “…humankind is known to have evolved on a diet with a ratio of n-6 to n-3 FA of close to 1:1. However, Western societies today are characterized by increased n-6 and trans FA and decreased n-3 FA intakes... In other words, the current nutritional environment in developed countries largely differs from that on which our genetic patterns were established roughly forty millennia ago... Therefore, a significant reduction in n-6 FA is desirable because excessive doses of n-6 PUFA and extremely high n-6 to n-3 FA ratios (i.e., 15–17:1) promote the pathogenesis of a wide range of chronic diseases, including inflammatory bowel disease, rheumatoid arthritis, cancer, and cardiovascular and autoimmune diseases; whereas decreased n-6:n-3 ratios have been shown to exert suppressive effects” [Toth et al. (2019) Int. Dairy J., 99, 104552].

Response: Thank you very much for this valuable comment. As suggested by the Reviewer for a fragment of the effect of n-6 and n-3 PUFA on human health has been corrected.

The added part is as follows:

“Proportions of specific groups of fatty acids in products are of special importance from the nutritional perspective. Excessive amounts of n-6 polyunsaturated fatty acids (PUFA) and a very high n-6/n-3 ratio, as is found in today’s Western diets, promote the pathogenesis of many diseases, including cardiovascular disease, cancer, and inflammatory and autoimmune diseases, whereas increased levels of n-3 PUFA (a low n-6/n-3 ratio) exert suppressive effects [45-48].”

Reviewer 2 Report

I think that this study is only of moderate interest as numerous studies already exist on the fatty acid profile of dairy products and especially cheese. The only novel aspect is the focus on Polish cheese from different milks.  However as you state these cheeses are quite varied in terms of production and therefore many factors influencing the fatty acid  profile cannot be elucidated.

You have presented the results in terms of milk source of the cheese and I can understand why, but realistically the make procedures for some different types of cheese can be so different that in theory that influence may even be greater than the milk source and thus this is not captured in anyway.

One aspect that i felt needed much more explanation was the methodology used to determine the total fatty acid content. The IDF method was referenced but a critical sub point in this method is the use of a basic or acidic catalysed esterification procedure. It is recommended for good reason in products where the free fatty acid content is high (likely the case with some of your cheeses) that the acidic catalysed approach is used. Otherwise you are very likely to underestimate the total fatty acid content as you fail to esterify many free fatty acids (as per the catalysed approach). Many researchers fail to appreciate the importance of fat extraction and the gas chromatography side but this can easily lead to incorrect conclusions.

I would also have issues with only doing this analysis in duplicate as triplicate analysis is standard for both total and free fatty analysis of dairy samples due to the big differences in values, fat content and presence of other compounds that can influence the extraction.

The manuscript contains many grammatical errors  that require addressing.

There is no need to state "analysed cheeses" throughout the manuscript.

I also think that you should state more optimal lipid quality indices for each value not just some.

Excessive number of references for a research study.

Line 62 : technological value - what does this mean?

Line 65/66: An important compound which influences the nutritional value and sensory attributes of cheese is milk fat. Please restructure sentence but also provide a reference

Line 89 : filtered, how?

Lines 91-95: This is unclear please restructure and clarify

In the Results and Discussion section most of the text is just statements of values without providing sufficient information as to their relevance. I think this aspects needs to be completely rewritten and the significance of some of the findings is not clear.

I also think that the conclusion does not fully reflect the outcome of the study.

Some minor mistakes are evident in the Reference section.

Author Response

Response to Reviewer 2 Comments

Thank you very much for  reviews of the manuscript. Please find below the answers to the suggested remarks. All changes have been included in the manuscript.

Reviewer:  The conclusions of the research are missing from the Abstract

Response: As suggested by the Reviewer, Abstract has been corrected.

Reviewer:  One aspect that i felt needed much more explanation was the methodology used to determine the total fatty acid content. The IDF method was referenced but a critical sub point in this method is the use of a basic or acidic catalysed esterification procedure. It is recommended for good reason in products where the free fatty acid content is high (likely the case with some of your cheeses) that the acidic catalysed approach is used. Otherwise you are very likely to underestimate the total fatty acid content as you fail to esterify many free fatty acids (as per the catalysed approach). Many researchers fail to appreciate the importance of fat extraction and the gas chromatography side but this can easily lead to incorrect conclusions.

Response: Thank you very much for this comment. I agree that the methylation method and the chromatographic separation conditions influence the results on fatty acid profile. The IDF method was used for methylation because, according to its scope, it is recommended for the preparation of methyl esters of milk fat and fat obtained from dairy products. According to literature data, methylation with the addition of KOH in methanol has been used by various authors, including: Seckin et al. (2005), Donmez et al. (2005), Prandini et al. (2007), Żegarska et al. (2008), Cossignani et al. (2014), Hirigoyen et al. (2018).

Reviewer:  The manuscript contains many grammatical errors  that require addressing.

Response: As suggested by the Reviewer, the manuscript has been corrected by a professional language editor.

Reviewer: There is no need to state "analysed cheeses" throughout the manuscript.

Response: As suggested by the Reviewer, the manuscript has been corrected.

Reviewer: Excessive number of references for a research study.

Response: As suggested by the Reviewer, this has been corrected.

Reviewer: Line 62 : technological value - what does this mean?

Response: As suggested by the Reviewer, this has been corrected. Lines 78-80 have been added: “Small differences in the composition of various milks may cause differences in the nutritional value and selected traits of the technological suitability of milk in the dairy industry [Barłowska, J. et al., Nutritional value and technological suitability of milk from various animal species used for dairy production. Comprehensive Reviews in Food Science and Food Safety, 2011, 20, 291-302].”

Reviewer: Line 65/66: An important compound which influences the nutritional value and sensory attributes of cheese is milk fat. Please restructure sentence but also provide a reference.

Response: As suggested by the Reviewer, this has been corrected. Lines 85-86 have been added: “The nutritional and sensory values of cheeses are influenced by many factors, including milk type and composition, starter cultures, and cheese-making technology [Papetti, P.; Carelli, A. Composition and sensory analysis for quality evaluation of a typical Italian cheese: Influence of Ripening Period. Czech J. Food Sci. 2013, 31, 5, 438–444].”

Reviewer: Line 89 : filtered, how?

Response: Line 124 has been added: “was filtrated by paper filter with anhydrous sodium sulphate”

Reviewer: Lines 91-95: This is unclear please restructure and clarify.

Response: As suggested by the Reviewer, this part has been changed.

Reviewer: In the Results and Discussion section most of the text is just statements of values without providing sufficient information as to their relevance. I think this aspects needs to be completely rewritten and the significance of some of the findings is not clear.

Response: As suggested by the Reviewer, the section “Results and Discussion” has been completed.

Reviewer: I also think that the conclusion does not fully reflect the outcome of the study.

Response: As suggested by the Reviewer, the conclusion has been corrected.

Round 2

Reviewer 1 Report

The overall quality of the manuscript has been largely improved. I have no further comments or suggestions.

Author Response

2 Response to Reviewer 2 Comments

Thank you very much for  reviews of the manuscript. Thank you for all the comments.

Reviewer 2 Report

The style and grammer is much improved. I still have concerns about the total fatty acid esterification method used, as it appears that you have used the basic approach. I realize that it is a ISO/IDF method that has been widely utilised but it is not fool proof.  However, based on the fact that you followed a referenced approach i can accept it but please provide some more details without the reader having to check the references.

Author Response

2 Response to Reviewer 2 Comments

Thank you very much for  reviews of the manuscript. Thank you for all the comments. All changes have been included in the manuscript.

Reviewer:  The style and grammar is much improved. I still have concerns about the total fatty acid esterification method used, as it appears that you have used the basic approach. I realize that it is a ISO/IDF method that has been widely utilised but it is not fool proof.  However, based on the fact that you followed a referenced approach i can accept it but please provide some more details without the reader having to check the references.

Response: As suggested by the Reviewer has been corrected. Lines 136 - 143 have been added:

For this purpose, 50mg of extracted fat was placed inside a sealed ampule of a capacity of 7 mL, 2.5 mL hexane was then added to dissolve it. The ampoule content was shaken vigorously until the fat was completely dissolved. Next, 0.1 mL 2M methanolic KOH solution was added. The ampoule was vigorously mixed in the vortex mixer for 1 minute and then left for 5 more minutes at room temperature (ca. 20 °C). After that time, 0.25 g of sodium hydrogensulphate-monohydrate was added and then the mixture was spun for 3 minutes (approximately 3000 spins/minute). The top layer of prepared methyl esters was taken for gas chromatographic (GC) analysis.

The other remarks noted by the Reviewer have been corrected.

Lines 18-19 “In turn” has been changed to “The”

Lines 44-45 “of various saturation degrees” has been changed to “of various degree's of saturation”

Lines 149-150 mistakes have been corrected

Line 202 “Fatty Acid Composition and Lipid Quality Indices in fat from the analysed cheeses” has been changed to “Fatty Acid Composition and Lipid Quality Indices in cheeses”

Lines 255-256 “content was determined” has been changed to “but”
